# Comprehensive Analysis of Chronic Low Back Pain: Morphological and Functional Impairments, Physical Activity Patterns, and Epidemiology in a German Population-Based Cross-Sectional Study

**DOI:** 10.3390/bioengineering12080878

**Published:** 2025-08-14

**Authors:** Bernhard Ulrich Hoehl, Nima Taheri, Lukas Schönnagel, Luis Alexander Becker, Lukas Mödl, Sandra Reitmaier, Matthias Pumberger, Hendrik Schmidt

**Affiliations:** 1Center for Musculoskeletal Surgery, Charité—Universitätsmedizin Berlin, 10117 Berlin, Germany; 2Julius Wolff Institute, Berlin Institute of Health, Charité—Universitätsmedizin Berlin, 10117 Berlin, Germany; 3Institute of Biometry and Clinical Epidemiology, Charité—Universitätsmedizin Berlin, 10117 Berlin, Germany

**Keywords:** chronic, intermittent, low back pain, LBP, morphologic, functional, impairment

## Abstract

Low back pain (LBP) is the leading cause of disability worldwide. While studies often focus on the relationship between magnetic resonance imaging (MRI) findings and symptoms or the link between pain and disability, comprehensive assessments that incorporate both structural and functional impairments are lacking. This study prospectively includes standardized questionnaires, medical histories, clinical exams, and lumbar–pelvic MRI. Participants were grouped by pain status, physical activity, structural impairments (e.g., Pfirrmann, Krämer, Fujiwara, Meyerding), and posture/mobility deviations. Data were analyzed using the Kruskal–Wallis test. Of the 1262 participants, 392 (31%) reported chronic low back pain (cLBP), 226 (18%) had intermittent low back pain (iLBP), and 335 (27%) were pain-free. Significant differences were observed in high physical activity levels based on WHO criteria (cLBP: 79%, iLBP: 78%, no-BP(2): 86%, *p* = 0.020, η^2^ = 0.008). Morphological impairments were more prevalent in cLBP (75%) and iLBP (76%) compared to no-BP(2) (55%) (*p* = 0.000, η^2^ = 0.043). Functional impairments showed similar patterns (cLBP: 42%, iLBP: 51%, no-BP(2): 38%, *p* = 0.014, η^2^ = 0.010). Participants with functional impairments tended to be younger. Consequently, the current classification system for diagnostics needs to incorporate alternative categories to more accurately differentiate types of back pain, which could enhance therapeutic outcomes.

## 1. Introduction

Low back pain (LBP) is a leading contributor to age-standardized years lived with disability (YLD) globally, affecting both men and women [1]. The lifetime prevalence of LBP ranges between 49% and 70% [2], with recurrent episodes occurring in 20% to 44% of individuals within a single year, particularly within the working population [3]. Chronic LBP (cLBP), defined as pain persisting beyond 12 weeks, affects 5% to 10% of those initially experiencing LBP [4]. The Global Burden of Disease Study 2021 shows a significant increase in individuals suffering from LBP since 1990. The age-standardized prevalence of LBP is the highest in central and eastern Europe and Australasia and the lowest in East Asia, Andean Latin America, and Southeast Asia. The impact of age on case numbers increases with advancing age, and peaking occurred within the 50–54 age group [5].

Management strategies for LBP encompass up to 50 clinical practice guidelines [6], with over 5000 trials evaluating various therapies and their modifications. However, many of these therapies have failed to replicate claimed effects, leading to ongoing modifications [7]. This may be attributed to inadequate application of evidence-based strategies or insufficient evidence supporting the effectiveness of commonly employed treatments [8,9]. Many clinicians argue that subgrouping LBP is crucial for improving individualized therapy [10]. The multidimensional nature of the condition presents a clinical challenge, requiring that each contributing factor be individually assessed and appropriately prioritized [11]. Effective daily management can be resource-intensive and may necessitate improved infrastructure and the integration of modern technologies [12]. Therefore, there is a pressing need for more precise and objective diagnostic assessments to tailor treatment plans to individual biological factors [13].

In 90% of LBP cases, no specific pathophysiological cause can be identified [2]. Potential sources of pain include lumbar structures such as intervertebral discs, facet joints, or spondylolisthesis. However, clinical tests often fail to consistently pinpoint these structures as the source of pain [14]. Magnetic resonance imaging (MRI) has become increasingly prevalent in recent years, revealing more morphological abnormalities in lumbar structures among LBP patients [15,16]. Yet, many imaging findings are also common in asymptomatic individuals, raising questions about their diagnostic relevance [17]. Additionally, while there is debate over the relationship between LBP and morphological changes, a correlation with restricted movement appears plausible, as a limited range of motion (ROM) is rarely observed in LBP patients [18,19,20]. Recent reviews suggest an inverse relationship between LBP and physical activity, though definitions of physical activity vary, including total activity, leisure time activities, and intensity [21,22]. The overall health benefits of physical activity are well-established, with the World Health Organization (WHO) recommending 75–150 min of physical activity per week, depending on intensity [23].

Despite extensive research on the relationships between LBP and MRI findings, mobility, and physical activity, comprehensive analyses that simultaneously examine all three factors are lacking. This study aims to provide a detailed analysis of morphological and functional impairments in relation to physical activity levels to better understand the multifactorial nature of LBP and identify key factors contributing to its persistence. The specific objectives of the study are as follows:

To characterize structural and functional impairments of the spine/back and levels of functional activity in individuals with and without LBP.To analyze demographic factors (age, sex, Body Mass Index (BMI)) and clinical characteristics (pain intensity, pain duration).

## 2. Materials and Methods

### 2.1. Study Design

The ongoing “Berlin Back Study” is a prospective cross-sectional investigation registered with the German Clinical Trial Register (DRKS-ID: DRKS00027907), scheduled from 1 January 2022 to 31 December 2025. Recruitment of participants is conducted through multiple channels, including local promotion at Charité-Universitätsmedizin Berlin (via mailed flyers, notice boards, online platforms, and social media), public outreach (including newspapers, magazines, podcasts, and television), collaborations with local businesses and administrative bodies, and word-of-mouth referrals.

The study protocol adheres to the ethical principles outlined in the Helsinki Declaration [24] and has received approval from the Ethics Committee of Charité-Universitätsmedizin Berlin (approval numbers: EA1/058/21). Written informed consent has been obtained from all participants. The research follows the Strengthening the Reporting of Observational Studies in Epidemiology (STROBE) guidelines [25]. Data collection commenced on 1 January 2022, with the analysis scheduled to conclude on 24 April 2024, at a university hospital research center.

### 2.2. Study Participants

To be eligible for participation in the study, individuals were required to meet the following inclusion criteria: provision of written informed consent; being either asymptomatic (with no history of back, pelvis, or hip pain, and no prior spinal surgery) or symptomatic with cLBP; and being aged 18–72 years with a minimum of 12 weeks of daily pain.

Exclusion criteria included the following: professional, competitive, or elite athletes; individuals with acute infections; substance abuse; pregnancy; a BMI exceeding 28 kg/m^2^; central or peripheral neurological impairments (such as spinal cord injury, radicular symptoms, or sensory deficits); irritated, inflamed, or infected tissues in the back measurement areas; spinal fractures; osteoporosis; tumors or bone metastases; previous spinal surgery; current use of strong medications (e.g., antiepileptics, long-acting antihistamines, systemic glucocorticoids, immunosuppressive drugs); rheumatic diseases; active systemic diseases (including tuberculosis, collagenosis, multiple sclerosis, autoimmune diseases, acquired immune deficiency syndrome); internal conditions that could pose risks during measurements (such as coronary heart disease, heart failure, severe hypertension, chronic obstructive pulmonary disease); and malformations or anomalies of the lower extremities (e.g., knee or hip arthroplasty, arthrodesis).

### 2.3. Quantitative Variables and Data Collection

The study coordinator provided participants with detailed instructions regarding the study protocol and facilitated their progression through the following stages: (1) completion of questionnaires; (2) a clinical physical examination; and (3) assessment of back morphology and motion. These evaluations were conducted on the same day, with a total duration averaging 90 min. Additionally, participants had the option to undergo MRI and gait analysis within a 14-day period following the initial assessments.

### 2.4. Questionnaires

Participants primarily completed the questionnaires digitally using a survey program specifically developed for the study. The data collection was conducted under consistent conditions, including the same room and computer for all subjects at the study center. The standardized approach ensured uniformity in the completion of the questionnaires, which included measures of pain intensity and duration, pain-related disability (based on Von Korff et al., 1992 [26]), and physical activity levels assessed using the International Physical Activity Questionnaire (IPAQ) [27].

### 2.5. Physical Examination

The physical examination, conducted by a skilled orthopedic consultant, encompassed an assessment of organ function; overall patient impression; and vital signs including respiratory rate, heart rate, and blood pressure. Neurological status was evaluated through tests of coordination, reflexes, sensitivity, and motor function.

Functional parameters of the lumbar spine and pelvis, such as posture, shape, orientation, and movement, were assessed using established clinical tests (e.g., Ott and Schober test, 3-step hyperextension test, passive lumbar extension test) and participant self-assessment. Measurements, recorded with precision (e.g., distances in centimeters, angles in degrees), focused on pain provocation. Self-reported functional limitations were rated on a scale from 1 (least severe) to 10 (most severe).

Pain characteristics, including location, type, course, radiation, intensity, quality, and duration, were detailed, along with exacerbating or relieving factors, triggers, and patient-reported causes. The medical history covered previous illnesses of the spine, surgeries of the lower extremity, pain history, treatments, treatment effects, and medication use. Family and social history included occupational status, family medical history of back pain, and stressful life circumstances, as well as past or current use of addictive substances such as alcohol and nicotine. Demographic data, including age, sex, body height, weight, hip diameter, and waist diameter, were also recorded.

### 2.6. Back Morphology and Motion Analysis

All participants underwent back shape measurements in both the sagittal and frontal planes while standing and sitting, using the Idiag M360^®^ (MediMouse, Idiag AG, Fehraltorf, Switzerland). This device measures segmental angles of the thoracic and lumbar spine. Participants were assessed in various postures—upright, flexed, extended, and left and right lateral bending—both in standing and sitting positions, with each position held for approximately 10 s and repeated three times. For standing measurements, maximum upper body flexion, extension, and lateral bending were performed with knees extended and arms crossed during extension to avoid interfering with the measurement. The sequence of tasks was randomized. Measurements were conducted by trained medical students, and the validity and reliability of the SpinalMouse have been confirmed in previous studies [28,29,30,31,32,33].

### 2.7. Spino-Pelvic MRI

MRI scans were performed using a 1.5 Tesla MRI scanner (Philips, Hamburg, Germany) with the following sequences: (1) Sagittal T1 (4 mm slices); (2) Sagittal T2 (4 mm slices); (3) Coronal STIR-T2 (4 mm slices); and (4) Axial T2 (3 mm slices). These scans were analyzed for various spinal pathologies, including intervertebral disc degeneration (Pfirrmann classification; [34,35]), disc herniation (Kramer classification; [35,36]), facet joint arthrosis (Fujiwara classification; [37]), spondylolisthesis (Meyerding classification; [38]), osteochondrosis intervertebralis (Modic classification; [39]), and spinal canal stenosis (Schizas classification; [40]) at each lumbar spine level.

The MRI assessments were conducted independently and blindly by three experienced investigators. In cases of discrepancies, the median classification was used.

### 2.8. Definition of Pain Status

Participants were categorized into three groups: chronic low back pain (cLBP), intermittent low back pain (iLBP), and no back pain (no-BP(2)). To determine these classifications, participants reported their current and past back pain experiences via questionnaires. During the clinical examination, a skilled orthopedic consultant verified the self-reported pain status, adhering to the definition of daily back pain lasting at least 12 weeks. Pain localization was documented through a detailed examination and classified into primary and secondary areas. Participants with daily lower back pain lasting 12 weeks or more were classified into the cLBP group. In contrast, participants who had experienced low back pain for less than 12 weeks and/or in intermittent episodes were classified as having intermittent low back pain (iLBP). Individuals without any history of back pain were assigned to the no-BP(2) group.

### 2.9. Characterization of Pain Status by Subgroups

Participants were systematically assigned to 24 subgroups based on morphological and functional parameters, physical activity levels, and pain status (Table 1). A participant was classified as morphologically or functionally impaired if any associated parameter was classified as impaired. For example, to be assigned to groups i, v, ix, xiii, xvii, or xxi, both morphological and functional assessments must be classified as non-impaired (Table 1).

#### 2.9.1. Morphological Impairments

MRI Analysis: Intervertebral discs and facet joints were graded using established degeneration scores. According to Pfirrmann et al., discs were categorized into five grades: grades 1 and 2 indicated no to minor changes and were considered non-impaired, while grades 3 to 5, reflecting moderate to severe degeneration, were deemed impaired [34,35]. Facet joints were evaluated based on Krämer et al.’s classification, with grades 0 to 2 indicating no to minor changes (non-impaired) and grades 3 to 5 indicating significant degeneration (impaired) [35,41]. Facet joint degeneration was also classified into four grades by Fujiwara et al., where grades 1 and 2 were non-impaired and grades 3 and 4 indicated impairment [37]. Spondylolisthesis was assessed using Meyerding’s classification, with grade 1 (spondylolisthesis ≤25%) considered non-impaired and grades 2 to 5 (spondylolisthesis >25%) considered impaired [38].Back Morphology Measurements: Segmental angles of the lumbar spine in an upright standing position were compared to a reference database of healthy individuals. Participants exhibiting deviations beyond two standard deviations from the norm were classified as impaired.

#### 2.9.2. Functional Impairments

Segmental angles of the lumbar spine during flexion, extension, and lateral bending (left and right) in a standing position were compared with a reference database of healthy individuals. Participants with deviations exceeding two standard deviations from the norm in any posture were considered functionally impaired.

#### 2.9.3. Level of Physical Activity

Participants’ physical activity levels were retrospectively categorized according to the WHO guidelines, which recommend a minimum of 150 min of moderate-intensity or 75 min of vigorous-intensity physical activity per week, or an equivalent combination. Meeting this criterion was classified as high physical activity. Vigorous physical activity was defined as activities such as aerobics, running, fast cycling, or fast swimming performed for at least 10 consecutive minutes. Moderate physical activity included activities such as carrying light loads, cycling at a regular pace, or swimming at a regular pace, also for a minimum of 10 uninterrupted minutes. Walking was not classified as either moderate or vigorous activity [23]. This assessment was conducted using the IPAQ.

### 2.10. Analysis of Demographic and Clinical Characteristics

To gain deeper insights into back pain, demographic data (age, sex, BMI), and clinical characteristics were visualized using boxplots, violin plots, and bar plots. Clinical characteristics included chronic pain grade and characteristic pain intensity, which was derived from current pain, the worst pain in the past three months, and average pain over the past three months, along with pain duration. Group differences were assessed using the non-parametric Kruskal–Wallis test to account for potential non-normal distributions.

### 2.11. Statistical Analysis

Descriptive analyses were conducted using boxplots, violin plots, bar, and column plots. Group differences were assessed using the non-parametric Kruskal–Wallis test. Effect sizes (Kruskal–Wallis η^2^) were interpreted based on standard thresholds: 0.01 to <0.06 for small effects, 0.06 to <0.14 for moderate effects, and ≥0.14 for large effects [42,43]. The significance level was set at *p* = 0.05. All statistical analyses were performed using R software [44] and RStudio software Version 2023.12.1.402 [45].

## 3. Results

### 3.1. Study Population

From January 2022 to April 2024, a total of 1262 participants were examined. The presented analysis on LBP enrolled 335 participants without back pain (male = 140, female = 195), 226 participants with iLBP (male = 104, female = 122), and 392 participants with cLBP (male = 172, female = 220). The remaining 309 participants were not included in this analysis due to back pain without involvement of the lower back. A significant but small age difference was observed between the groups (*p* = 0.001, Kruskal–Wallis η^2^ = 0.014), with median ages of 39 years (IQR: 29–52) for the no-BP(2) group, 40 years (IQR: 32–52) for the iLBP group, and 44 years (IQR: 34–53) for the cLBP group. No significant differences were found between groups in terms of sex (*p* = 0.609) or BMI (*p* = 0.805). Within the cLBP group, 40 (10.2%) participants had a history of lower extremity surgery, compared to 25 (11.1%) participants in the iLBP group and 23 (6.9%) participants in the no-BP(2) group (*p* = 0.183, η^2^ = 0.004).

### 3.2. Characterization of Pain Status to Subgroups

#### 3.2.1. Morphological Impairment

The classification of lumbar spine morphological changes was based on the Pfirrmann, Krämer, Fujiwara, and Meyerding grading systems. Significant differences between the pain groups were observed in all segments, except for the Meyerding classification and in the L1/2 and L2/3 segments of the Fujiwara classification (Figure 1). Five participants presented with six lumbar vertebrae. Lumbar spine alignment in the standing position showed significant but small differences between the pain status groups (*p* = 0.001, η^2^ = 0.015). Median lumbar spine alignment was −25° (IQR: −31° to −19°) for the cLBP group, −24° (IQR: −28° to −19°) for the iLBP group, and −26° (IQR: −31° to −21°) for the no-BP(2) group. Overall, significant differences in morphological impairments were observed between the groups (*p* < 0.001, η^2^ = 0.043), with 75% of cLBP, 76% of iLBP, and 55% of no-BP(2) participants showing impairments.

#### 3.2.2. Functional Impairment

In basic functional tests assessing lumbar spine mobility while standing, significant differences were observed between participants without back pain and those with back pain. Mobility in extension (η^2^ = 0.022, *p* < 0.001), right lateral bending (η^2^ = 0.027, *p* < 0.001), and left lateral bending (η^2^ = 0.011, *p* = 0.007) were notably different in the no-BP(2) group in comparison to the groups of cLBP and iLBP. The mobility cut-off values for 90% of no-BP(2) participants were as follows: flexion (8.9° to 36°), extension (−48° to −20°), right lateral bending (−29° to −1°), and left lateral bending (9° to 31°) (Figure 2). Overall, functional impairments differed significantly between pain groups (*p* = 0.014, η^2^ = 0.010), with 42% of cLBP, 51% of iLBP, and 38% of no-BP(2) participants exhibiting functional impairments.

#### 3.2.3. Physical Activity

The metabolic equivalent of task (MET) minutes, as assessed by the IPAQ, did not reveal significant differences between the pain groups (cLBP: median = 2837, IQR = 1555–4513; iLBP: median = 3099, IQR = 1673–5088; no-BP(2): median = 3092, IQR = 1710–4971 [all in MET minutes]; η^2^ = 0.006, *p* = 0.052). Based on the WHO classification, a significantly higher proportion of participants without back pain demonstrated high levels of physical activity (cLBP: 79%, iLBP: 78%, no-BP(2): 86%; η^2^ = 0.008, *p* = 0.020) (Figure 3).

#### 3.2.4. Assigning Participants to Subgroups

A total of 10% of study participants with either low or high physical activity levels are pain-free and exhibit no morphological or functional impairments. Among cLBP patients, 8.7% (low activity) and 5.9% (high activity) show no morphological or functional impairments. Interestingly, the pain-free groups exhibit both morphological and functional impairments in 7.2% (low activity) and 7.9% (high activity) of cases, which is only slightly different from the iLBP groups but significantly and clinically lower compared to the cLBP groups, with 13.8% and 11% respectively (Table 2).

### 3.3. Impact of Demographic and Clinical Characteristics on Subgroup Differentiation

There were no significant differences between sexes in subgroups with low physical activity (no-BP(2): η^2^ = 0.018, *p* = 0.877; iLBP: η^2^ = 0.194, *p* = 0.078; cLBP: η^2^ = 0.084, *p* = 0.152) or high physical activity (no-BP(2): η^2^ = 0.032, *p* = 0.065; iLBP: η^2^ = 0.031, *p* = 0.244; cLBP: η^2^ = 0.002, *p* = 0.924). Participants with a high level of physical activity and functional impairment only (groups xv, xix, xxiii) were significantly younger (no-BP(2): η^2^ = 0.108, *p* < 0.001; iLBP: η^2^ = 0.144, *p* < 0.001; cLBP: η^2^ = 0.059, *p* = 0.924). Body Mass Index (BMI) displayed significant differences among groups i–iv (η^2^ = 0.268, *p* = 0.022) (Figure 4).

### 3.4. Impact of Physical Activity on Subgroup Differentiation

For participants with low levels of physical activity, there were no significant differences in chronic pain grade (η^2^ = 0.061, *p* = 0.551), characteristic pain intensity (η^2^ = 0.138, *p* = 0.064), or duration of pain (η^2^ = 0.120, *p* = 0.264) between those with iLBP and cLBP. Within each pain status group, no significant differences were observed between participants with no morphological or functional impairment (Table 2, groups v, ix), morphological impairment only (Table 2, groups vi, x), functional impairment only (Table 2, groups vii, xi), or both morphological and functional impairment (Table 2, groups viii, xii).

For participants with high levels of physical activity, there were significant differences in chronic pain grade (η^2^ = 0.058, *p* = 0.003) and characteristic pain intensity (η^2^ = 0.146, *p* < 0.001) between those with iLBP and cLBP. However, there was no significant difference in pain duration (η^2^ = 0.015, *p* = 0.763) between these groups. Within each pain status group, no significant differences were observed between participants with no morphological or functional impairment (Table 2, groups xvii, xxi), morphological impairment only (groups xviii, xxii), functional impairment only (Table 2, groups xix, xxiii), or both morphological and functional impairment (Table 2, groups xx, xxiv) (Figure 5).

## 4. Discussion

This study examines the influence of physical activity levels, morphological, and functional impairments on the subclassification of LBP into cLBP, iLBP, and no-BP(2) categories. Our findings reveal that 10% of participants across both low and high activity levels were pain-free and displayed no morphological or functional impairments. Notably, cLBP patients showed a lower percentage of individuals without impairments (8.7% for low activity and 5.9% for high activity) compared to pain-free participants, who exhibited a slightly higher incidence of both morphological and functional impairments.

For high physical activity participants, significant differences were observed in chronic pain grade (η^2^ = 0.058, *p* = 0.003) and characteristic pain intensity (η^2^ = 0.146, *p* < 0.001) between iLBP and cLBP groups, although no significant difference in pain duration was found. This suggests that high physical activity influences pain severity and intensity but not necessarily the duration of pain.

Our results further indicate that a significant number of participants with high levels of physical activity were pain-free and exhibited fewer impairments overall. According to the WHO classification, there were significantly more participants with high physical activity among those without back pain (cLBP: 79%; iLBP: 78%; no-BP(2): 86%; *p* = 0.020, η^2^ = 0.008). The percentage of morphological impairments was significantly different between pain statuses, with the highest prevalence in cLBP (75%) and iLBP (76%) compared to no-BP(2) (55%; η^2^ = 0.043; *p* < 0.001). Functional impairments also varied significantly, being more prevalent in cLBP (42%) and iLBP (51%) than in no-BP(2) (38%; η^2^ = 0.010; *p* = 0.014).

The proposed subclassification revealed significant age differences, with participants having functional impairments being the youngest in all pain status groups (cLBP: η^2^ = 0.072, *p* < 0.001; iLBP: η^2^ = 0.126, *p* < 0.001; no-BP(2): η^2^ = 0.109, *p* < 0.001). However, within each pain status, there was no significant correlation between functional or morphological impairments and chronic pain grade, pain intensity, or pain duration. This suggests that while age and physical activity levels influence impairment and pain status, the specific impact of these factors on pain severity remains complex.

Our study’s results are consistent with other large-scale studies regarding the physical activity levels of participants. Notably, our participants reported a median of approximately 3000 MET minutes per week, which is higher than the 1000 MET minutes reported in other studies [21]. This suggests that our voluntary participant group is particularly active.

Interestingly, our findings on BMI and age align with the existing literature. We observed that morphologic impairments tended to increase with age and were marginally associated with higher BMI, although no significant differences in BMI were found across pain statuses (*p* = 0.805). This consistency with the literature [17,46] reinforces the importance of considering age and BMI in the context of LBP.

The MRI findings of our study indicated a tendency for morphologic impairments to increase towards the lower segments, particularly involving vertebral disc degeneration (Pfirrmann and Krämer classification in Figure 1), aligning with the body of literature [47]. In our study, in the majority of lumbar segments, the grades of degeneration were associated with pain status. Existing studies show inconsistent correlations between disc degeneration and LBP [48,49], which may be attributed to different characteristics of the study populations [50]. The extent to which subclassification can be refined by integrating further clinical parameters remains unclear. Recent studies have begun to investigate genetic predisposition as a means to deepen our understanding of low back pain [51].

In our study population, functional impairment showed a statistically significant (*p* = 0.014) but weak correlation with pain status (η^2^ = 0.010). Despite the statistical significance, the clinical relevance remains questionable, particularly given the absence of an established minimal clinically important difference for spinal ROM. Based on the distribution of spinal ROM presented in Figure 2 (Section 3.2.2. Functional Impairment), it appears unlikely that a clinician could reliably assign a patient to a specific pain category based solely on movement amplitude. In our study population, the clinical impact on ROM may be further attenuated due to the relatively high levels of physical activity and low BMI among participants. This finding is consistent with the previous literature; the systematic review and meta-analysis by Errabity et al. demonstrated significant reductions in lumbar ROM across sagittal, frontal, and transverse planes, although these findings were not uniform across all studies [18].

Several limitations must be considered. The cross-sectional design limits our ability to infer causality and assess changes over time. Additionally, our voluntary participant sample may not be fully representative, and the exclusion of individuals with a BMI over 28 kg/m^2^ or previous surgeries could introduce bias towards a healthier population. The subjective nature of pain also introduces variability in the data. In addition, comorbidities (e.g., diabetes, high blood pressure) possibly influencing the results were assessed.

Future research should focus on prospective interventional studies to explore the effectiveness of individualized treatments based on physical activity levels and impairment types. Improved subclassification of LBP can potentially enhance therapeutic strategies and outcomes.

Our study contributes to the understanding of LBP by incorporating individual characteristics, such as physical activity and impairments, into the diagnosis. Although age and BMI showed significant differences between subgroups, a clinically meaningful subclassification based solely on these or other individual parameters remains challenging.

Musculoskeletal health professionals should remain aware of the multifactorial nature of LBP. A thorough, interdisciplinary assessment that includes lifestyle, spinal morphology, and functional capacity is essential to address patient-specific needs effectively. Future research is needed to assess the impact of these characteristics on treatment efficacy and to refine LBP management strategies.

## Figures and Tables

**Figure 1 bioengineering-12-00878-f001:**
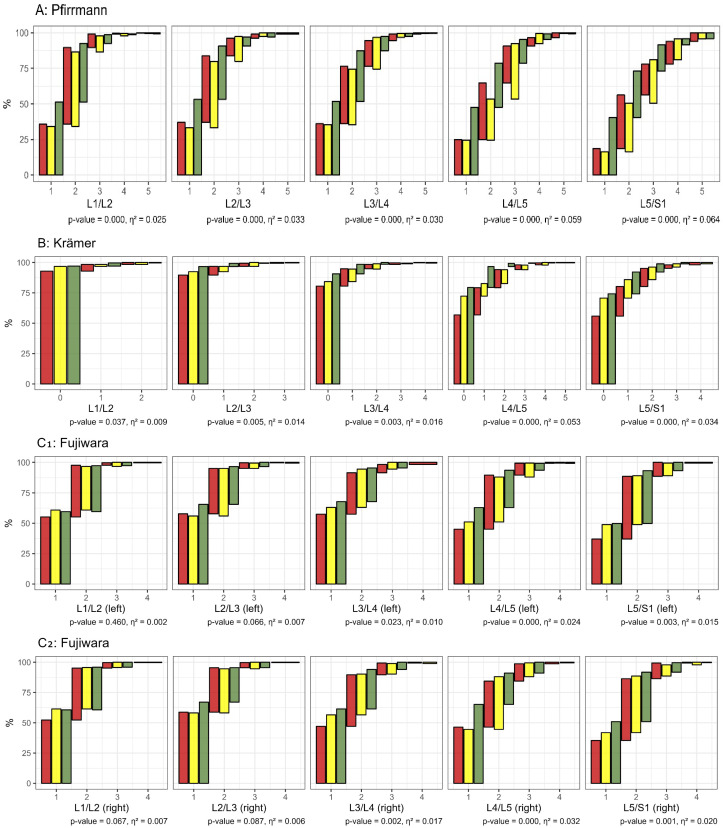
Comparison of the morphologic changes of the lumbar spine classified by Pfirrmann (disc degeneration, (**A**); Krämer (disc herniation, (**B**); Fujiwara (facet joint degeneration, left: (**C_1_**); right: (**C_2_**) and Meyerding (spondylolisthesis, (**D**) of the group with chronic low back pain (cLBP, red) and intermittent low back pain (iLBP, yellow) and without back pain (no-BP(2), green).

**Figure 2 bioengineering-12-00878-f002:**
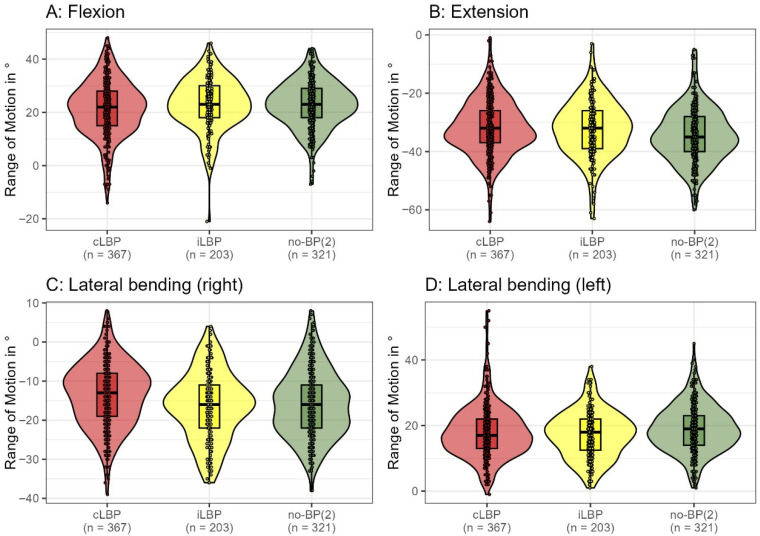
Lumbar spine mobility assessed using the Idiag M360^®^(MediMouse, Idiag AG, Fehraltorf, Switzerland), across three primary groups: chronic low back pain (cLBP), intermittent low back pain (iLBP), and pain-free individuals (no-BP(2)). Significant intergroup differences were found in extension (η^2^ = 0.022, *p* < 0.001), right lateral bending (η^2^ = 0.027, *p* < 0.001), and left lateral bending (η^2^ = 0.011, *p* = 0.007), indicating reduced mobility in specific movements among those with LBP.

**Figure 3 bioengineering-12-00878-f003:**
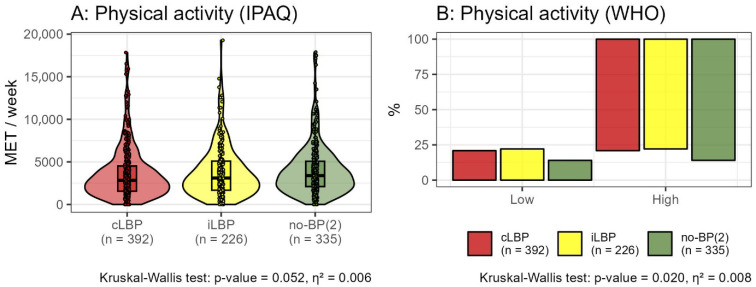
Level of physical activity, measured in metabolic equivalent (MET) minutes per week as assessed by the International Physical Activity Questionnaire (IPAQ) (**A**). IPAQ shows no significant difference (*p* = 0.052). However, according to World Health Organization (WHO) classifications (**B**), there are significant differences in physical activity levels among participants with chronic low back pain (cLBP), intermittent low back pain (iLBP), and those without back pain (no-BP(2)). Participants are classified as having a high level of physical activity if they engage in either 150 min of moderate-intensity physical activity or 75 min of vigorous-intensity physical activity per week.

**Figure 4 bioengineering-12-00878-f004:**
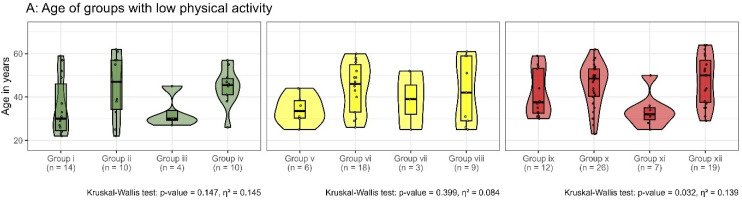
Age (**A**,**B**) and Body Mass Index (BMI) (**C**,**D**) are presented for participants categorized by their pain status: without back pain (green), with intermittent low back pain (yellow), and with chronic low back pain (red). Within each color-coded group, the four subsets represent no morphological or functional impairment (first), morphological impairment only (second), functional impairment only (third), and both morphological and functional impairment (fourth). Participants are classified as having a high level of physical activity (**B**,**D**) if they engage in either 150 min of moderate-intensity physical activity or 75 min of vigorous-intensity physical activity per week. Differences within each individual graph are evaluated using the Kruskal–Wallis test.

**Figure 5 bioengineering-12-00878-f005:**
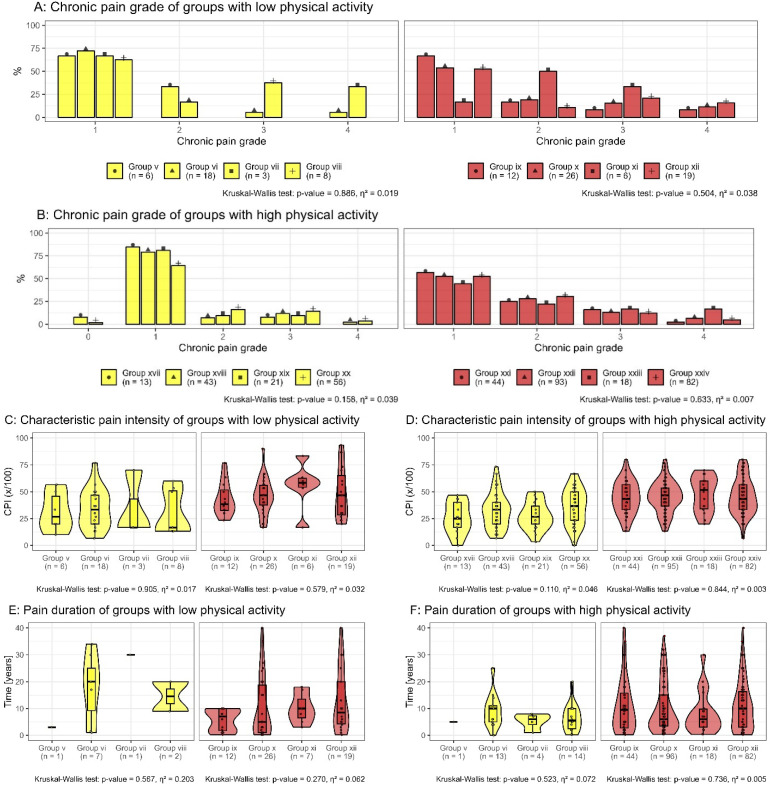
The chronic pain grade (**A**,**B**) and characteristic pain intensity (**C**,**D**) according to Korff et al. among pain duration (**E**,**F**) are displayed for the subgroups of intermittent low back pain (yellow) and chronic low back pain (red). In the four-pack of each color, the first group had neither morphologic nor functional impairment, the second one had morphologic impairment only, the third one had functional impairment only, and the fourth one had both morphologic and functional impairment. The participants are considered to have a high level of physical activity (**B**,**D**,**F**) when they spend either 150 min of moderate physical activity or 75 min of vigorous physical activity per week. The Kruskal–Wallis test checks for differences within each individual graph.

**Table 1 bioengineering-12-00878-t001:** Assignment of study participants into the subgroups based on their pain status, physical activity, and functional or morphologic impairment: **✓**: not impaired, **✗**: impaired, no-BP(2) = no back pain, iLBP = intermittent low back pain, cLBP = chronic low back pain.

**Low Physical Activity**	**no-BP(2)**		**Group i**	**Group ii**	**Group iii**	**Group iv**
		Morphology	** ✓ **	** ✗ **	** ✓ **	** ✗ **
		Function	** ✓ **	** ✓ **	** ✗ **	** ✗ **
	**iLBP**		**Group v**	**Group vi**	**Group vii**	**Group viii**
		Morphology	** ✓ **	** ✗ **	** ✓ **	** ✗ **
		Function	** ✓ **	** ✓ **	** ✗ **	** ✗ **
	**cLBP**		**Group ix**	**Group x**	**Group xi**	**Group xii**
		Morphology	** ✓ **	** ✗ **	** ✓ **	** ✗ **
		Function	** ✓ **	** ✓ **	** ✗ **	** ✗ **
**High Physical Activity**	**no-BP(2)**		**Group xiii**	**Group xiv**	**Group xv**	**Group xvi**
		Morphology	** ✓ **	** ✗ **	** ✓ **	** ✗ **
		Function	** ✓ **	** ✓ **	** ✗ **	** ✗ **
	**iLBP**		**Group xvii**	**Group xviii**	**Group xix**	**Group xx**
		Morphology	** ✓ **	** ✗ **	** ✓ **	** ✗ **
		Function	** ✓ **	** ✓ **	** ✗ **	** ✗ **
	**cLBP**		**Group xxi**	**Group xxii**	**Group xxiii**	**Group xxiv**
		Morphology	** ✓ **	** ✗ **	** ✓ **	** ✗ **
		Function	** ✓ **	** ✓ **	** ✗ **	** ✗ **

**Table 2 bioengineering-12-00878-t002:** Assignment of study participants with no back pain (no-BP(2)), intermittent low back pain (iLBP), and chronic low back pain (cLBP) into the subgroups (i)–(xxiv): Each table cell indicates the number of participants (*n*), number of sexes specified in brackets (males, females), the proportion (%) relative to the corresponding activity group, and the proportion (%) relative to of the whole group (excluding the group with missing data).

**Low Physical Activity**	**no-BP(2)** ***n* = 47** **(17, 30)** **(4.9%)**	**Group i**	**Group ii**	**Group iii**	**Group iv**	**Missing Data**
		*n* = 14 (6, 8) 10%/1.9%	*n* = 10 (4, 6) 7.2%/1.3%	*n* = 4 (1, 3) 2.9%/0.5%	*n* = 10 (3, 7) 7.2%/1.3%	*n* = 9 (3, 6)
	**iLBP** ***n* = 50** **(18, 32)** **(5.2%)**	**Group v**	**Group vi**	**Group vii**	**Group viii**	
		*n* = 6 (2, 4) 4.3% / 0.8%	*n* = 18 (4, 14) 13% / 2.4%	*n* = 3 (0, 3) 2.1% / 0.4%	*n* = 9 (6, 3) 6.5%/1.2%	*n* = 14 (6, 8)
	**cLBP** ***n* = 82** **(38, 44)** **(8.6%)**	**Group ix**	**Group x**	**Group xi**	**Group xii**	
		*n* = 12 (7, 5) 8.7%/1.6%	*n* = 26 (12, 14) 19%/13.5%	*n* = 7 (1, 6) 5.1%/0.9%	*n* = 19 (12, 7) 13.8%/2.6%	*n* = 18 (6, 12)
*n* = 138 (19% of the entire group) + 41 with missing data (MRI or SpinalMouse)						
**High Physical Activity**	**no-BP(2)** ***n* = 288** **(123, 165)** **(30%)**	**Group xiii**	**Group xiv**	**Group xv**	**Group xvi**	**Missing Data**
		*n* = 74 (40, 34) 12%/10%	*n* = 65 (22, 43) 11%/8.7%	*n* = 30 (15, 15) 4.9%/4%	*n* = 59 (22, 37) 9.7%/7.9%	*n* = 60 (24, 36)
	**iLBP** ***n* = 176** **(86, 90)** **(18%)**	**Group xvii**	**Group xviii**	**Group xix**	**Group xx**	
		*n* = 13 (8, 5) 2.1%/1.7%	*n* = 45 (18, 27) 7.4%/6%	*n* = 21 (10, 11) 3.5% / 2.8%	*n* = 58 (34, 24) 9.6%/7.8%	*n* = 39 (16, 23)
	**cLBP** ***n* = 310** **(134, 176)** **(33%)**	**Group xxi**	**Group xxii**	**Group xxiii**	**Group xxiv**	
		*n* = 44 (20, 24) 7.2%/5.9%	*n* = 97 (39, 58) 16%/13%	*n* = 18 (7, 11) 3%/2.4%	*n* = 83 (36, 47) 14%/11%	*n* = 68 (32, 36)
*n* = 607 (81% of the entire group) + 167 with missing data (MRI or SpinalMouse)						

## Data Availability

The data that support the findings of this study are available from the corresponding author, B.U.H., upon reasonable request.

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
