# Peer review of "Comprehensive Analysis of Chronic Low Back Pain: Morphological and Functional Impairments, Physical Activity Patterns, and Epidemiology in a German Population-Based Cross-Sectional Study"

_bioengineering, 2025, doi:10.3390/bioengineering12080878_

Round 1

Reviewer 1 Report

Comments and Suggestions for Authors

Comprehensive Analysis of Chronic Low Back Pain: Morphological and Functional Impairments, Physical Activity Patterns, and Epidemiology in a German Population-Based Cross-Sectional Study

Hoehl BU et al

iU bioengineering, MDPI

General.

This nice and well performed propective study with a proper design was performed on patients with chronic  and intermittent LBP, because  comprehensive assessments that incorporate both structural and functional impairments are lacking. Therefore authors combined standardized questionnaires, medical histories, clinical exams, and lumbar-pelvic MRI to provide a detailed analysis of morphological and functional impairments in relation to physical activity levels to better understand the multifactorial nature of LBP and identify key factors contributing to its persistence.

Authors present variation in high physical activity between groups and similar groupwise variation in functional impairments and, but did not try to associate between MRI findings, activity levels and functional impairments. This could have been an added value, since it is well known, so far,  that imaging findings and non specific chronic or intermittent low back pain are poorly related to eachother. Authors state that the current classification system for diagnostics needs to incorporate alternative categories. Authors also state that many clinicians argue that subgrouping LBP is crucial for improving individualized therapy. This might be so true, however, to my opinion the work of the authors does not provide concrete direction to alternative classification categories that could help, while meanwhile a lot of research has been started to study all kinds of biomarkers to get better insights in the transition of acute to chronic pain, addressing amongst others immunology, inflammation, epigenetics etc. Although I appreciate authors’ nice and well performed work, I wonder if going on with characterizing impairments, clinical findings  and demographics will provide beter insights. To my opinion the authors’ work offer too little novel value, therefore I hesitate to recommend this work for publication.

SPECIFIC COMMENTS

Abstract

Authors present numbers for cLBP, iLBP en individuals being pain-free, together 76% of the included population. What about the other 24%?

What is (2) in no-BP(2)? What does it mean?

P5, 181 Participants with daily lower back pain lasting 12 weeks or more were classified into the cLBP group, those with lower back pain in recurrent episodes of less than 12 weeks were classified into the iLBP group independently of the time passed since the first occurence…..

I assume that patients with an LBP episode shorter than 12 weeks also should have had a minimum of one more LBP episodes in the past. What are criteria for iLBP inclusion? I recommend to concretely mention this.  

P6, 221 Level of Physical Activity. Please mention in this section the main criteria with respect to categories low and high physical activity according to IPAQ

P9 Figure 1. I recommend to add what test is measuring what at top of the figure or in the legends. This makes it better understandable for the less familiar readers. By example: Fujiwara (facet joint degeneration)

P9, 276. Overall, functional impairments differed significantly between pain groups (p = 0.014, .² = 0.010), with 42% of cLBP, 51% of iLBP, and 38% of no-BP(2) participants exhibiting functional impairment.

Although statistically significant these numbers do not show great clinical differences. What do the authors think? Should be addressed in the discussion section.

P11, 307 ….which is only slightly different from the iLBP groups but significantly higher compared to the cLBP groups, with13.8% and 11% respectively (Table 2).

Should significant ‘lower’ be better replaced by significant ‘higher’? And I suggest to add ‘clinically‘ to significant

P11, Table 2. Please add more explanation to this table. Usually numbers of individuals in groups are presented with n=… And for Group i (17,30): what do these numbers mean? F/M? This applies also for 10% / 1.9% and for all cells in table 2. All figures and tables should be well readible and understandable in itself. I suggest to provide appropriate information for all content in tables and figures.

It is not clear to me whether participants classified as iCLBP were self-reported pain free at the moment of inclusion or not, and suffered less than 12 weeks at moment of inclusion ánd had former LBP episodes. Can authors explain? If patients suffered from back pain or not, this might be of influence on the test results. Can authors reply to this?

Could authors have made an attempt to associate findings with respect to MRI, functional impairments and activity to each other? This would be of more interest. Because the results presented in it selves do to my opinion marginally provide new information.

P14, 359, According to the WHO classification, there were significantly more participants with high physical activity among those without back pain (cLBP: 79%; iLBP: 78%; no-BP(2): 86%; p = 0.020, η² = 0.008). The percentage of morphological impairments was significantly different between pain statuses, with the highest prevalence in cLBP (75%) and iLBP (76%) compared to no-BP(2) (55%; .² = 0.043; p < 0.001). Functional impairments also varied significantly, being more prevalent in cLBP (42%) and iLBP (51%) than in no-BP(2) (38%; .² = 0.010; p = 0.014).

Although ther are statistical significant differences, clinical results differ slightly. Maybe less than we could expect. Can authors comment on this?

Discussion section: I think the data could be more elaborated on and discussed in this section. A bit more depth is welcome and may enrich the the significance of the findings.

Author Response

Dear Reviwer,

thank you very much for your detailed review and thoughtful suggestions.

Comment 1:
Authors present numbers for cLBP, iLBP en individuals being pain-free, together 76% of the included population. What about the other 24%?

Response 1:
Thank you for pointing this out. As the manuscript focuses on LBP, we included participants with different states of LBP (iLBP, cLBP, pain-free). In the detailed clinical physical examination, the remaining 24% were found to have back pain localized outside the lower back (e.g., thoracic spine). I have added this information on page 7, line 244, in the section “3.1. Study population”.

Comment 2:
What is (2) in no-BP(2)? What does it mean?

Response 2:
The symbol (2) refers to the two key questions—one from the questionnaire and one from the physical examination—in which participants indicated they had ‘no back pain.’ Although this level of detail may not be strictly necessary for the present analysis, this study is part of a large, multi-year, and multidisciplinary research project. Therefore, we opted for consistent terminology throughout to ensure that readers can clearly associate findings across different publications.

Comment 3:
P5, 181 Participants with daily lower back pain lasting 12 weeks or more were classified into the cLBP group, those with lower back pain in recurrent episodes of less than 12 weeks were classified into the iLBP group independently of the time passed since the first occurence…..

I assume that patients with an LBP episode shorter than 12 weeks also should have had a minimum of one more LBP episodes in the past. What are criteria for iLBP inclusion? I recommend to concretely mention this.  

Response 3:
Thank you for your comment — I understand your reasoning. To avoid dividing the participants into too many subgroups, the inclusion criterion was defined as any number of low back pain (LBP) episodes, each lasting less than 12 weeks. I clarified the definition on page 5, line 181, in the section '2.8. Definition of pain status'.

Comment 4:
P6, 221 Level of Physical Activity. Please mention in this section the main criteria with respect to categories low and high physical activity according to IPAQ

Response 4:
Thank you for the comment. I revised the corresponding section (2.9.3. Level of Physical Activity).

Comment 5:
P9 Figure 1. I recommend to add what test is measuring what at top of the figure or in the legends. This makes it better understandable for the less familiar readers. By example: Fujiwara (facet joint degeneration)

Response 5:
Thank you for pointing this out. I added the information to the legends of Figure 1.

Comment 6:
P9, 276. Overall, functional impairments differed significantly between pain groups (p = 0.014, .² = 0.010), with 42% of cLBP, 51% of iLBP, and 38% of no-BP(2) participants exhibiting functional impairment.

Although statistically significant these numbers do not show great clinical differences. What do the authors think? Should be addressed in the discussion section.

Response 6:
Thank you for the comment. I agree that the clinical difference on the range of motion solely is limited. I addressed it in the discussion (p 14, L411).

Comment 7:
P11, 307 ….which is only slightly different from the iLBP groups but significantly higher compared to the cLBP groups, with13.8% and 11% respectively (Table 2).

Should significant ‘lower’ be better replaced by significant ‘higher’? And I suggest to add ‘clinically‘ to significant

Response 7:
Thank you for pointing this out. We agree and I changed it accordingly (P10, 316, section 3.2.4. Assigning participants to subgroups)

Comment 8:
P11, Table 2. Please add more explanation to this table. Usually numbers of individuals in groups are presented with n=… And for Group i (17,30): what do these numbers mean? F/M? This applies also for 10% / 1.9% and for all cells in table 2. All figures and tables should be well readible and understandable in itself. I suggest to provide appropriate information for all content in tables and figures.

Response 8:
Thank you for your valuable feedback, I agree. I have expanded the explanation in table 2, P10, L 320.

Comment 9:
It is not clear to me whether participants classified as iCLBP were self-reported pain free at the moment of inclusion or not, and suffered less than 12 weeks at moment of inclusion ánd had former LBP episodes. Can authors explain? If patients suffered from back pain or not, this might be of influence on the test results. Can authors reply to this?

Response 9:
Thank you for your valuable comment. We fully understand the concern regarding the potential influence of pain at the exact time of examination.

Participants categorized as having chronic low back pain (cLBP) reported experiencing low back pain on a daily basis for at least 12 weeks prior to the study assessment. However, due to the heterogeneous clinical presentation of cLBP—including persistent pain with mild or strong fluctuations, intermittent pain attacks with pain-free intervals, or continuous pain at varying intensity levels—it is possible that participants experienced periods (<24 hours; there must be pain every day) of reduced or absent symptoms. As a result, the 90-minute physical examination may have coincided with such a low-pain or even pain-free moment, despite the overall chronic pain classification.
The inclusion criterion for intermittent low back pain (iLBP) was defined as any number of low back pain (LBP) episodes, each lasting less than 12 weeks. This means that participants with iLBP may have been experiencing an active pain episode at the time of examination, or their most recent episode may have already resolved. Consequently, the 90-minute physical examination could have been conducted during a pain-free period, even in individuals with a history of recurrent low back pain.
To account for the participants’ current pain status (in both cLBP and iLBP groups), various established pain-related instruments were administered and analyzed, including the Chronic Pain Grade and Characteristic Pain Intensity (incorporating the Numeric Rating Scale, NRS).

Comment 10:
Could authors have made an attempt to associate findings with respect to MRI, functional impairments and activity to each other? This would be of more interest. Because the results presented in it selves do to my opinion marginally provide new information.

Response 10:
Thank you very much for this important comment. We fully agree that the interrelations among MRI findings, functional impairments, and physical activity are of great clinical and scientific interest. However, several studies have already investigated specific associations in detail, such as between disc degeneration and pain status.

In clinical reality, patients often present with a complex constellation of imaging findings, varying degrees of functional impairment, and different levels of physical activity. To reflect this real-world complexity, the aim of our study was not to explore a single specific association in depth, but rather to provide a comprehensive overview by integrating multiple dimensions. We therefore examined the broader pattern of interactions by including spinal morphology (e.g., disc degeneration, disc herniation, facet joint degeneration, spondylolisthesis, and posture assessed via segmental angles in upright standing), spinal function (flexion, extension, lateral flexion), and physical activity, in relation to various pain characteristics (pain status, intensity, duration, and disability), while adjusting for relevant demographic factors (age, BMI).

Moreover, our study benefits from a unique control group of healthy individuals who underwent MRI despite having no history of back pain or other complaints that would normally justify imaging. This allowed for a more differentiated comparison,  and enhances the clinical relevance of the findings.

Comment 11:
P14, 359, According to the WHO classification, there were significantly more participants with high physical activity among those without back pain (cLBP: 79%; iLBP: 78%; no-BP(2): 86%; p = 0.020, η² = 0.008). The percentage of morphological impairments was significantly different between pain statuses, with the highest prevalence in cLBP (75%) and iLBP (76%) compared to no-BP(2) (55%; .² = 0.043; p < 0.001). Functional impairments also varied significantly, being more prevalent in cLBP (42%) and iLBP (51%) than in no-BP(2) (38%; .² = 0.010; p = 0.014).

Although ther are statistical significant differences, clinical results differ slightly. Maybe less than we could expect. Can authors comment on this?

Response 11:
Thank you very much for your valuable comment. We agree. The well-known phenomenon that spinal degeneration can also be present in asymptomatic individuals appears to be more pronounced in our study. One possible explanation may lie in the recruitment process: our participants were drawn from the general population, excluding individuals with prior spinal surgery and with a limited BMI range. In contrast, the study populations in many comparable investigations are often derived from inpatient or surgical settings, which may inherently involve more severe cases. We expanded the discussion concerning morphological and functional impairments (P 14/15, L 402-422)

Comment 12:
Discussion section: I think the data could be more elaborated on and discussed in this section. A bit more depth is welcome and may enrich the the significance of the findings.

Response 12:
Thank you for this comment. I agree and we have expanded the “Discussion” section in several aspects.

Reviewer 2 Report

Comments and Suggestions for Authors

I found the article well-written and interesting.
From a purely formal standpoint, the article is well-structured: the sections (abstract, introduction, materials and methods, and results) are well-structured and well-written. There are no problems with the English language and the tables and figures help clarify the content.
From a content standpoint, the research question is ambitious but sufficiently narrow, and the research design seems appropriate. The authors aimed to analyze chronic back pain through a cross-sectional, population-based study in Germany. Their goal was to explore the correlations between pain, morphological and functional alterations of the spine, and levels of physical activity. The sample is very large, and the inclusion and exclusion criteria are well-explained. All ethical procedures (informed consent, ethics committee) were properly followed. Participants were divided into groups based on the presence of pain, level of physical activity, and the presence of morphological and functional alterations. The results are well explained and stated in scientifically accurate, non-emphatic tones, just as the limitations and future research perspectives are well presented.
The literature review is robust but could be further enriched by including even more recent studies (2023-2025), particularly regarding the effectiveness of classification and treatment personalization strategies in patients with low back pain, given the rapidly evolving evidence in the fields of musculoskeletal and personalized medicine. If it is not possible to include even more recent studies, a justification from the authors regarding their motivation would be helpful.

Author Response

I found the article well-written and interesting.
From a purely formal standpoint, the article is well-structured: the sections (abstract, introduction, materials and methods, and results) are well-structured and well-written. There are no problems with the English language and the tables and figures help clarify the content.
From a content standpoint, the research question is ambitious but sufficiently narrow, and the research design seems appropriate. The authors aimed to analyze chronic back pain through a cross-sectional, population-based study in Germany. Their goal was to explore the correlations between pain, morphological and functional alterations of the spine, and levels of physical activity. The sample is very large, and the inclusion and exclusion criteria are well-explained. All ethical procedures (informed consent, ethics committee) were properly followed. Participants were divided into groups based on the presence of pain, level of physical activity, and the presence of morphological and functional alterations. The results are well explained and stated in scientifically accurate, non-emphatic tones, just as the limitations and future research perspectives are well presented.
The literature review is robust but could be further enriched by including even more recent studies (2023-2025), particularly regarding the effectiveness of classification and treatment personalization strategies in patients with low back pain, given the rapidly evolving evidence in the fields of musculoskeletal and personalized medicine. If it is not possible to include even more recent studies, a justification from the authors regarding their motivation would be helpful.

Response:
We would like to sincerely thank the reviewer for the thoughtful and encouraging feedback. We appreciate the recognition of our manuscript.
Regarding your helpful suggestion to further enrich the literature review with more recent studies from 2023–2025, we have now updated the introduction and discussion sections accordingly. We addressed current developments in the fields of musculoskeletal and personalized medicine on P2 L 52.
Thank you again for your constructive and positive review.

Reviewer 3 Report

Comments and Suggestions for Authors An interesting work, a subject that hasn't been fully explored or fully explained so far. However, I have a few comments: ABSTRACT: Well written. Please explain the abbreviation MRI. Introduction: Well written, extensive literature review. Please describe the epidemiology of LBP in more detail, depending on patient age and geographic distribution.

Materials and methods: Well written. Please add detailed patient descriptions (BMI, gender distribution). Please describe whether comorbidities (diabetes, hypertension, degenerative changes, post-spinal and lower limb injuries) were assessed – these factors could have influenced the results. Good quality tables and figures. Results: Good quality tables and figures. Please add a breakdown by comorbidities. DISCUSSION: Discussion too short. Please expand the discussion. Please describe your results by comorbidities (diabetes, hypertension, degenerative changes, post-spinal and lower limb injuries) – these factors could have influenced the results. Please describe the limitations of the study. CONCLUSIONS: Please add 2-3 practical conclusions for orthopedists and rehabilitation specialists.

Author Response

An interesting work, a subject that hasn't been fully explored or fully explained so far. However, I have a few comments: 

Thank you for your review and the recognition of our manuscript.

Comment 1:
ABSTRACT: Well written. Please explain the abbreviation MRI. 

Response 1:
We agree and added the explanation on P1, L16.

Comment 2:
Introduction: Well written, extensive literature review. Please describe the epidemiology of LBP in more detail, depending on patient age and geographic distribution.

Response 2:
Thank you for the comment. I added a more detailed epidemiology of LBP on P1, L 41.

Comment 3:
Materials and methods: Well written. Please add detailed patient descriptions (BMI, gender distribution).

Response 3:
Thank you very much for your comment. We apologize for any inconvenience. We reported the patient characteristics (gender distribution, age and BMI) in the Results section P7, L254, section “3.1. Study population”. A detailed analysis of age and BMI on subgroup differentiation is shown in the Section “3.3. Impact of demographic and clinical characteristics on subgroup differentiation” (P 11, L329).

Comment 4:
Please describe whether comorbidities (diabetes, hypertension, degenerative changes, post-spinal and lower limb injuries) were assessed – these factors could have influenced the results. Good quality tables and figures. Results: Good quality tables and figures. Please add a breakdown by comorbidities. DISCUSSION: Discussion too short. Please expand the discussion. Please describe your results by comorbidities (diabetes, hypertension, degenerative changes, post-spinal and lower limb injuries) – these factors could have influenced the results. Please describe the limitations of the study. 

Response 4:
Thank you very much for your valuable and thoughtful comment. I fully agree that comorbidities such as diabetes may influence both pain perception and potentially spinal degeneration. Unfortunately, internal medical comorbidities were not systematically documented in the current study. However, spinal degeneration was assessed in detail using MRI (P8, L 266, section “3.2.1. Morphological impairment”). Participants with spinal fractures, tumors, or a history of spinal surgery were excluded from the study. Previous surgeries of the lower extremities were recorded.
I specified the content of the physical examination (P 4, L 146, section “2.5. Physical examination”) accordingly.
I added the numbers of previous operations on the  lower extremity (no significant difference) (P7, L262, section “3.1. Study population”).
I expanded the limitation of the study (especially missing comorbidities) (P 15, L 425).
We expanded the “Discussion” section in several aspects.

Comment 5:
CONCLUSIONS: Please add 2-3 practical conclusions for orthopedists and rehabilitation specialists.

Response 5:
Thank you for pointing this out. I modified the conclusion accordingly (P15 L 432).

Round 2

Reviewer 3 Report

Comments and Suggestions for Authors The authors revised the manuscript based on the comments. The manuscript is acceptable in its current form.